# Thermotherapy Plus Neck Stabilization Exercise for Chronic Nonspecific Neck Pain in Elderly: A Single-Blinded Randomized Controlled Trial

**DOI:** 10.3390/ijerph17155572

**Published:** 2020-08-01

**Authors:** Ho-Jin Shin, Sung-Hyeon Kim, Suk-Chan Hahm, Hwi-Young Cho

**Affiliations:** 1Department of Health Science, Gachon University Graduate School, Incheon 21936, Korea; sports0911@hanmail.net (H.-J.S.); 315201@hanmail.net (S.-H.K.); 2Graduate School of Integrative Medicine, CHA University, Seongnam 13488, Korea; 3Department of Physical Therapy, Gachon University, Incheon 21936, Korea

**Keywords:** neck stabilization exercise, nonspecific neck pain, salt pack, thermotherapy

## Abstract

Neck pain is a serious problem for public health. This study aimed to compare the effects of thermotherapy plus neck stabilization exercise versus neck stabilization exercise alone on pain, neck disability, muscle properties, and alignment of the neck and shoulder in the elderly with chronic nonspecific neck pain. This study is a single-blinded randomized controlled trial. Thirty-five individuals with chronic nonspecific neck pain were randomly allocated to intervention (*n* = 18) or control (*n* = 17) groups. The intervention group received thermotherapy with a salt-pack for 30 min and performed a neck stabilization exercise for 40 min twice a day for 5 days (10 sessions). The control group performed a neck stabilization exercise at the same time points. Pain intensity, pain pressure threshold (PPT), neck disability index, muscle properties, and alignment of the neck and shoulder were evaluated before and after the intervention. Significant time and group interactions were observed for pain at rest (*p* < 0.001) and during movement (*p* < 0.001), and for PPT at the upper-trapezius (*p* < 0.001), levator-scapula (*p* = 0.003), and splenius-capitis (*p* = 0.001). The disability caused by neck pain also significantly changed between groups over time (*p* = 0.005). In comparison with the control group, the intervention group showed significant improvements in muscle properties for the upper-trapezius (tone, *p* = 0.021; stiffness, *p* = 0.017), levator-scapula (stiffness, *p* = 0.025; elasticity, *p* = 0.035), and splenius-capitis (stiffness, *p* = 0.012), and alignment of the neck (*p* = 0.016) and shoulder (*p* < 0.001) over time. These results recommend the clinical use of salt pack thermotherapy in addition to neck stabilization exercise as a complementary intervention for chronic nonspecific neck pain control.

## 1. Introduction

Neck pain is a common health problem with a lifetime prevalence of 14.2% to 71% in the adult population and is considered a major problem for public health [1]. In particular, Korean women of middle and older age have a prevalence of 20.8% [2]. The common presentation of neck pain is nonspecific neck pain, defined as simple neck pain without a specific underlying disease causing the pain, which results from postural and mechanical causes [3,4]. Appropriate management of nonspecific neck pain is essential because chronic neck pain results in increased muscle tone, restricted cervical range of motion, functional impairments of activities of daily living, and decreased quality of life [4].

Nonspecific neck pain can be treated with a variety of interventions, such as medication, manual therapy, heat, and exercise [3,4,5,6]. In particular, exercise is an evidence-based practice to not only relieve pain in individuals with nonspecific neck pain, but also to improve muscle strength, motor function, and quality of life [7]. The efficacy of cervical-scapulothoracic stabilization exercise and neck stabilization exercise for the management of neck pain have been reported in previous studies [8,9,10].

Thermotherapy has been used to reduce chronic musculoskeletal pain and has been reported as a complementary intervention [11,12,13,14,15,16,17,18,19]. Since the application of thermotherapy to the skin increases the temperature and blood flow to the muscle and decreases muscle fatigue [14,15,16], it may be associated with an increase in muscle flexibility [17]. These effects of thermotherapy can also decrease muscle spasms [13]. Considering these findings, the application of thermotherapy followed by exercise during the rehabilitation process may strengthen the stability of neck muscles; thus, thermotherapy combined with neck stabilization exercise may be more effective than exercise alone for relieving nonspecific neck pain.

A hot pack is one of the most common methods of thermotherapy, and various heat transfer substances, such as silicate gel, polymer gel, and water, were used in the hot pack [20,21,22,23]. Salt can be an option for a heat transfer substance in hot packs. Considering that thermotherapy using salt have analgesic and anti-inflammatory effects [24,25], hot packs using salt can be used for management of musculoskeletal pain. However, no clinical trial has been specifically conducted to investigate the feasibility of salt packs in patients with nonspecific neck pain, and the efficacy of thermotherapy combined with neck stabilization exercise for nonspecific neck pain has not been investigated. Thus, the aim of this study was to investigate the efficacy of a combination of a salt pack with neck stabilization exercise on pain, pain pressure threshold (PPT), neck disability, and alignment in individuals with chronic nonspecific neck pain. To this end, we compared the effects of thermotherapy using a salt pack plus neck stabilization exercise versus a neck stabilization exercise alone for symptomatic relief from chronic nonspecific neck pain.

## 2. Methods

### 2.1. Study Design

This study was designed as a single-blinded, randomized controlled trial. The experimental protocol was approved by the Gachon University Institutional Review Board (1044396-201903-HR-040-01). The study was performed in accordance with the protocol, and all participants provided written informed consent prior to their enrollment in the study.

### 2.2. Participants and Sample Size

For this study, we enrolled elders (>60 years) with chronic nonspecific neck pain that had lasted longer than 6 months (visual analogue scale (VAS) > 3/10), who had not undertaken regular physical activity in the past year. Chronic nonspecific neck pain was defined as neck pain provoked by neck postures, movements, or pressure for at least 3 months without a known pathology (neurological, trauma-induced, etc.) as the cause of the complaints [26]. The exclusion criteria were neck pain associated with inflammatory, hormonal, and neurological disorders or structural deformity in the upper extremities; neck pain related to previous surgery; positive radicular signs consistent with nerve root compression; severe referred pain; severe psychological disorder; or pregnancy. In addition, participants were excluded if they were under anti-inflammatory, analgesic, anticoagulant, muscle relaxant, or antidepressant medication use 1 week before the study commenced [26].

The sample size was calculated using the computer software G-power (Heinrich-Heine-University Düsseldorf, version 3.1.9.4, Düsseldorf, Germany) In the present study, the effect size was set to 0.25 (medium effect size) [27], and the alpha level was 0.05. On the basis of these values, 34 participants (17 participants per group) were needed to achieve 80% power using a 2-sided test. Thus, with a 10% dropout rate, a total of 38 participants were required.

### 2.3. Experimental Procedures and Interventions

All participants were randomly assigned to treatment or control groups using a stratified randomization method [28]. Participants were stratified by age (60–69/70–79) and baseline VAS at rest (3–5/6–8) and randomization was performed within each stratum by using permuted block randomization (block size 4). The group allocation was concealed to the outcome assessor by blinding the group assignment, and primary and secondary variables were assessed before and after the intervention. Pre-test were performed on the morning (9 a.m. to 10 a.m.) before the first intervention, and post-tests were performed in the morning (9 a.m. to 10 a.m.) on the next day after the intervention. All assessments were conducted in a random order to exclude potential fatigue and order effects due to measurement order.

The intervention group performed neck stabilization exercise and thermotherapy using a salt pack, and the control group performed only neck stabilization exercises at Saesum Resort in Taean-gun. The neck stabilization exercise was applied by slightly modifying the exercise intervention performed in the previous study [10]. It consisted of a warm-up (5 min), main exercise (30 min), and cool-down (5 min), and was performed in both the intervention and control groups. The warm-up and cool-down consisted of neck and upper extremity stretching, and the main exercise was as follows: (1) Deep neck flexor isometric exercise in supine position; (2) Multi-directional isometric exercise (cervical flexion, extension, rotation, side bending) in a sitting position; (3) Upper extremity movement exercise; (4) Resistive exercise with Thera-band. The neck stabilization exercise was performed according to the therapist’s instruction.

After the neck stabilization exercise, the intervention group performed additional thermotherapy using a salt pack. For thermotherapy, bay salt was used in packs. The salt was collected at Taean-gun, Chungcheongnam-do, Republic of Korea in April 2019 and then packed in cotton cloth. The far infrared radiation (FIR) emissivity of bay salt used in this study was 0.900 µm and the FIR emission power was 3.89 × 10^2^ W/m^2^·µm (Table A1 in Appendix A). The salt packs were kept in a warming cabinet (LH-1043G, Lassele Co., Ltd., Ansan, Korea) set at 60 °C until the start of the intervention. The participant was in a prone position. A salt pack set at 55 °C was applied to the neck and shoulder [29]; even after 30 min of application, it was maintained at about 40–50 °C. All interventions were conducted twice a day for 5 days, neck stabilization exercise was performed for 40 min and additional thermotherapy using salt pack was performed for 30 min.

### 2.4. Outcome Measures

#### 2.4.1. Primary Variables

The visual analogue scale (VAS) [30] was used to assess pain intensity at rest and during movement. VAS at rest (resting pain) was defined as an unpleasant feeling or pain without movement, and VAS during movement (movement-induced pain) was defined as unpleasant feelings or pain incurred by neck movement (flexion, extension, lateral flexion, rotation) [31]. Patients marked their pain intensity at rest and during movement on a VAS table.

To assess the PPT of the neck, the PPT assessment method described in a previous study was used with a distal algometer (Somedic AB, Farsta, Sweden) containing a 1-cm^2^ probe [11,32,33]. The pressure head of the algometer was applied to the upper trapezius, levator scapula, and splenius capitis of the neck and shoulder area, as in a previous study [34]. The assessor gradually increased the application pressure in 10-kPa/s increments until the participants expressed a pain response, such as a pain-induced vocalization and a gesture related to pain (hand grasp or eye blink) [32]. The measurement was repeated twice, and the measurement interval was 30 s. The mean threshold was calculated for the left- and right-side points.

The neck disability index (NDI) was used to assess functional disability due to neck pain; this assessment consists of 10 items describing the impact of pain on different daily living activities [33]. Each item is rated on a six-point Likert scale (range 0–5), with 0 indicating no limitation due to pain and 5 indicating that an activity is impossible to perform. The total score ranges from 0 to 50, with a higher score indicating a higher level of disability. The NDI is the most widely used tool for assessing functional outcomes in patients with neck pain and is recommended for evaluation of the effectiveness of neck pain treatment.

#### 2.4.2. Secondary Variables

A handheld myotonometer (Myoton AS, Tallinn, Estonia) with excellent intra and inter-tester reliability (ICC = 0.97) was used to measure the mechanical properties of muscle (muscle tone, stiffness, and elasticity) [35]. The skeletal muscle assessments were performed at the same region where PPT was measured. Each time, the probe (3 mm diameter) of equipment was placed perpendicular to the skin’s surface and five repeated measurements were obtained. The myofascial tissue oscillations were evoked with 5 brief (15 ms) mechanical impulses at 0.4 N force and frequency of 1 Hz. The mean threshold was calculated for the left- and right-side points. Muscle tone is a value expressing muscle tone in a passive or resting state without voluntary contraction. Muscle stiffness is a value representing the resistance of tissue to external mechanical impulse. Muscle elasticity is a value expressing the ability to recover to the initial shape after the disappearance of the external force of deformation.

Changes in cervical and shoulder alignments were assessed using the cervical angle and shoulder angle, respectively [36] (Figure A1). the cervical angle and shoulder angle were defined through three markers (tragus of ear, spinous process of the C7, acromion) attached to the participants’ anatomical landmarks. Images were collected by a 16-megapixel camera (SM-N976N, Samsung, Suwon, Korea) with an acromion height of 1.5 m, located perpendicular to the ground by a spirit level. The collected images were processed through the MATLAB (version 2019b, Mathworks, Inc., Natick, MA, USA). The cervical angle is formed when a line drawn from the tragus of the ear to the C7 vertebra intersects a horizontal line, and the shoulder angle is formed when a horizontal line passing through the lateral shoulder meets the line drawn from C7 to the lateral shoulder.

### 2.5. Statistical Analysis

Data analyses were performed using IBM SPSS Statistics 25.0. (IBM-SPSS Inc, Chicago, IL, USA) The statistician was blinded to group allocation for all analyses. An independent *t*-test and the χ^2^ test was performed in order to compare general characteristics between the two groups. Repeated measures ANOVA was used to analyze the changes in variables between groups over time and main effect comparisons were performed. Post-hoc analysis was performed through independent *t*-test and paired *t*-test using Bonferroni methods. A *p*-value of < 0.05 was considered statistically significant.

## 3. Results

### 3.1. Participant Characteristics

A total of 53 participants were recruited, all of whom were women. Fifteen individuals were excluded from participating; 13 did not meet the inclusion criteria and two declined to participate. Because individuals scheduled their problems, in the intervention group, one individual did not participate in the final assessment. In the control group, two individuals declined to participate after allocation. However, there were no complaints or dropout due to the intensity of the intervention except for those who were dropped out for the above reasons. A total of 35 patients completed the study. Figure 1 shows the participant flow through the enrollment, allocation, assessment, and analysis stages.

There were no significant differences between the intervention and control groups in terms of the general participant characteristics (age, height, weight, and body mass index) (Table 1). Additionally, there are no significant differences in the baseline values of the outcome variables assessed in this study between the two groups.

### 3.2. Primary Outcomes

#### 3.2.1. Pain Intensity

As shown in Table 2, compared to neck stabilization exercise alone, salt pack therapy combined with neck stabilization exercise significantly improved pain intensity over time at rest (*p* < 0.001) and during movement (*p* < 0.001). The intervention group showed a significantly decreased pain intensity at rest (*p* < 0.001) and during movement (*p* < 0.001) after the intervention. The control group also showed a significantly decreased pain intensity at rest (*p* = 0.009) and during movement (*p* = 0.001).

#### 3.2.2. Pain Pressure Threshold

A significant increase in the PPT for the upper trapezius (*p* = 0.002), levator scapula (*p* < 0.001), and splenius capitis (*p* < 0.001) was observed in the groups over time (Table 2). In comparison with the control, the intervention group showed a significant improvement in PPT for the upper trapezius (*p* = 0.002), levator scapula (*p* < 0.001), and splenius capitis (*p* = 0.001). Both thermotherapy with neck stabilization exercise (upper trapezius, *p* < 0.001; levator scapula, *p* < 0.001; splenius capitis, *p* < 0.001) and neck stabilization exercise alone (upper trapezius, *p* = 0.014; levator scapula, *p* = 0.018; splenius capitis, *p* = 0.015) significantly increased PPT after treatment.

#### 3.2.3. Neck Disability

Significant improvement in disability due to neck pain was observed in groups over time (*p* = 0.005, Table 2). Interestingly, salt pack with neck stabilization exercise yielded significant improvements in disability due to neck pain in comparison with the improvements obtained with neck stability exercise alone (*p* = 0.005). The intervention group showed significant increases in NDI scores (*p* < 0.001). However, the control group did not show a significant change in NDI after neck stability exercise.

### 3.3. Secondary Outcomes

#### 3.3.1. Muscle Properties

In comparison with the control group, the intervention group showed a significant improvement over time in muscle tone (upper trapezius, *p* = 0.021), stiffness (upper trapezius, *p* = 0.017; levator scapula, *p* = 0.025; splenius capitis, *p* = 0.012), and elasticity (levator scapula, *p* = 0.035) (Table 3). The intervention group also showed significant differences in muscle tone (upper trapezius, *p* < 0.001; levator scapula, *p* = 0.003; splenius capitis, *p* = 0.006), stiffness (upper trapezius, *p* < 0.001; levator scapula, *p* < 0.001; splenius capitis, *p* < 0.001), and elasticity (upper trapezius, *p* = 0.001; levator scapula, *p* < 0.001; splenius capitis, *p* = 0.001) after the intervention. However, the control group did not show significant improvements in muscle tone, stiffness, and elasticity after neck stabilization exercise.

#### 3.3.2. Cervical and Shoulder Alignments

In assessments of neck posture correction, cervical (*p* = 0.016) and shoulder (*p* < 0.001) alignments significantly improved in the groups over time (Table 4). As shown in Table 4, cervical (*p* < 0.001) and shoulder (*p* < 0.001) angles significantly improved after salt pack therapy with neck stabilization exercise. However, the control group did not show significant improvements in cervical and shoulder angles. Salt pack combined with neck stabilization exercise was not significantly more effective than neck stabilization exercise only.

## 4. Discussion

This study aimed to compare the effects of thermotherapy combined with neck stabilization exercise to those of neck stabilization exercise alone on chronic nonspecific neck pain. This study is the first investigation to demonstrate that 10 sessions of salt pack thermotherapy plus neck stabilization exercise provide benefits that are superior to those of neck stability exercise alone on pain intensity, PPT, neck disability, muscle properties, and body alignment in individuals with chronic nonspecific neck pain. These results may provide evidence to use salt pack therapy plus neck stabilization exercise as a complementary intervention for relief from nonspecific neck pain.

Previous studies have reported the effects of therapeutic exercise, including neck stabilization exercise, with or without thermotherapy on nonspecific musculoskeletal pain and disability [8,9,10,11,12,34,37,38,39]. Our study also demonstrated that both thermotherapy using a salt pack plus neck stabilization exercise and neck stabilization exercise alone had significant effects in reducing pain intensity, increasing PPT, and improving disability. Interestingly, in comparison with neck stabilization exercise alone, the intervention group also showed significantly better neck pain control. In the study by Cramor et al. [12] both the thermotherapy and non-thermotherapy groups received their usual medication and physical therapy regimens during the study period, with the thermotherapy group receiving thermotherapy using mud packs; their findings suggested that the additional thermotherapy significantly alleviated nonspecific neck pain. Thermotherapy has been shown to effectively alleviate pain and improve somatosensory function in individuals with chronic neck pain [12]. The results of previous studies that applied thermotherapy with exercise for low back pain control support our findings [11]. In addition to the thermal effect, it appears that there is also the effect of FIR emitted from the bay salt. FIR can provide pain control and increased blood flow [40]. This effect of FIR may contribute to pain reduction and changes in muscle characteristics. The superiority of the intervention group may be explained by a reduction in pain intensity [11,12,37] and improvement in muscle flexibility [41] as a result of thermotherapy prior to neck stabilization exercise. These changes in pain intensity and PPT may have resulted in the decreased neck disability evidenced by the NDI results.

This study showed significant time and group interactions of PPT, and both intervention and control groups showed significant improvements in PPT. Prior studies have also reported that thermotherapy has a greater influence on PPT in comparison with other treatments for chronic neck pain [38,42]. However, a previous study [12] reported no significant change in PPT after thermotherapy application. That study explained that with hyperalgesia pressure is maintained by central sensitization in patients with chronic neck pain [43] and that thermotherapy had no effect on central sensitization. The discrepancies between the findings of our study and that study may be attributable to the alteration of pain memories associated with central sensitization in patients with chronic musculoskeletal pain via exercise [44]. A previous study [9] reported significant improvement in the PPT on the middle point of the upper trapezius in patients with nonspecific neck pain after neck stabilization exercise, which supports our results for PPT.

This study also examined the changes in muscle properties of the neck/shoulder in both groups. The intervention group demonstrated significantly decreased muscle tone, stiffness and elasticity, but the control group did not show significant changes in muscle properties. Thermotherapy increases the temperature of and blood flow to the muscle and reduces muscle fatigue [14,15,16], which may decrease muscle tone, stiffness, and elasticity. In addition, significant recovery of these muscle properties and neck pain may be associated with the significant differences in the effects on cervical and shoulder alignment between the two groups. Previous studies have reported that the high tone of the upper trapezius is associated with the forward neck [45,46], and that increased tone and stiffness of the neck and shoulder muscles can be a major physical factor for neck pain [47,48]. Our results showed a significant reduction in tone and stiffness of the neck and shoulder muscles, neck pain, and forward neck and round shoulder in the intervention group. These results showed that changes in muscle characteristics due to thermotherapy combined with neck stabilization exercise had a significant effect on neck and shoulder alignment and neck pain.

In the intervention group of this study, the intervention time for one session is more than one hour, which may be burdensome to the body. The participant’s condition was continuously checked during and after the intervention, and there were no adverse symptoms such as pain, fatigue, and delayed onset muscle soreness. Moreover, there were no complaints about the interventions, and no participants dropped out due to problems with interventions. It seems that there was no problem because active intervention (neck stabilization exercise) was performed for only 40 min and then thermos-intervention was performed for 30 min.

There are some limitations to the present study. First, since the current study assessed the findings only after 10 sessions applied over 5 days, a thorough understanding of the effects of repeated thermotherapy with neck stability exercises over longer periods is necessary to evaluate the clinical use of salt pack interventions. Second, all participants in this study were women, even though sex was not an inclusion/exclusion criterion in this study. To obtain more generalizable conclusions relating to the efficacy of the salt pack combined with neck stabilization exercise for chronic nonspecific neck pain, further studies with suitable sex ratios may be needed. Third, although this feasibility study showed significant effects on pain intensity, PPT, muscle properties, and aliment in individuals with chronic nonspecific neck pain, the small sample size may limit the generalizability of these results.

## 5. Conclusions

According to the results of this study, salt pack thermotherapy combined with neck stabilization exercise is superior to neck stabilization exercise alone for chronic nonspecific neck pain control. However, to generalize the clinical use of this intervention for management of nonspecific neck pain, further studies with larger sample sizes and longer periods of application are needed.

## Figures and Tables

**Figure 1 ijerph-17-05572-f001:**
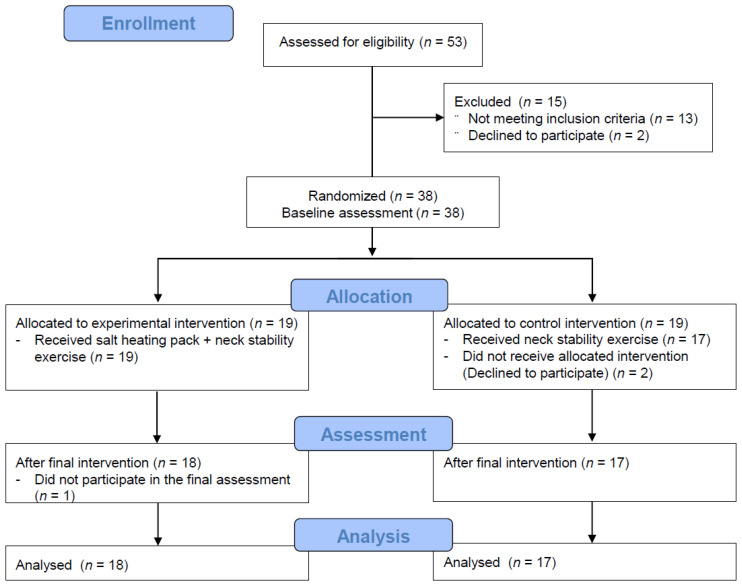
Flow diagram of study participants.

**Table 1 ijerph-17-05572-t001:** General characteristics of the participants.

Variable	Intervention Group (*n* = 18)	Control Group (*n* = 17)	P
Age (years)	68.06 ± 4.71	66.24 ± 4.71	0.261
Height (m)	1.54 ± 0.04	1.53 ± 0.03	0.299
Weight (kg)	58.92 ± 7.52	57.34 ± 4.68	0.457
BMI (kg/m^2^)	24.73 ± 2.93	24.5 ± 1.80	0.780
VAS at rest (cm)	4.78 ± 1.11	4.53 ± 1.37	0.560
Onset duration (month)	15.33 ± 7.76	14.29 ± 7.74	0.694
Job context ^†^			
Working	13 (72.22)	11 (64.71)	0.632
Non-working	5 (27.78)	6 (35.29)	

Data are expressed as mean ± standard deviation or number (%) ^†^. BMI, body mass index; P, *p*-value; VAS, visual analogue scale.

**Table 2 ijerph-17-05572-t002:** The changes in pain intensity, pain pressure threshold, and neck disability.

Outcome/Group	Baseline	Two Weeks Post-Treatment	P (Pairwise Comparison)	P (T * G)
**VAS at rest** **(cm)**				
Intervention group	4.78 ± 1.11	1.17 ± 1.04	<0.001	<0.001
Control group	4.53 ± 1.37	3.41 ± 1.28	0.009	
**VAS during movement** **(cm)**				
Intervention group	6.75 ± 1.06	2.28 ± 1.41	<0.001	<0.001
Control group	6.06 ± 1.14	4.53 ± 1.37	0.001	
**PPT_Upper trapezius (kg)**				
Intervention group	2.41 ± 0.50	4.28 ± 1.38	<0.001	0.002
Control group	2.56 ± 0.75	3.22 ± 0.87	0.014	
**PPT_Levator scapula (kg)**				
Intervention group	2.07 ± 0.51	4.12 ± 1.18	<0.001	<0.001
Control group	2.38 ± 0.89	2.99 ± 0.83	0.018	
**PPT_Splenius capitis (kg)**				
Intervention group	2.56 ± 0.93	4.52 ± 0.84	<0.001	0.001
Control group	2.90 ± 0.87	3.55 ± 0.78	0.015	
**NDI (%)**				
Intervention group	36.11 ± 12.88	16.56 ± 10.56	<0.001	0.005
Control group	33.65 ± 11.92	27.29 ± 10.79	0.052	

Data are expressed as mean ± standard deviation. P, *p*-value; T * G, time and group interaction; VAS, visual analog scale; PPT, pain pressure threshold; NDI, neck disability index.

**Table 3 ijerph-17-05572-t003:** The changes in muscle characteristics.

Outcome/Group.	Baseline	Two Weeks Post-Treatment	P (Pairwise Comparison)	P (T * G)
**Upper trapezius**				
**Tone (Hz)**				
Intervention group	13.71 ± 2.25	11.64 ± 0.91	<0.001	0.021
Control group	14.19 ± 2.72	13.88 ± 3.43	0.552	
**Stiffness (N/m)**				
Intervention group	255.56 ± 19.25	229.83 ± 27.64	<0.001	0.017
Control group	260.94 ± 11.33	254.88 ± 32.14	0.288	
**Elasticity (logarithm)**				
Intervention group	1.69 ± 0.26	1.97 ± 0.42	0.001	0.079
Control group	1.59 ± 0.23	1.67 ± 0.31	0.328	
**Levator scapula**				
**Tone (Hz)**				
Intervention group	19.03 ± 1.89	17.04 ± 2.23	0.003	0.129
Control group	20.11 ± 2.66	19.48 ± 3.10	0.331	
**Stiffness (N/m)**				
Intervention group	335.78 ± 48.50	280.06 ± 53.92	<0.001	0.025
Control group	345.00 ± 48.90	333.35 ± 54.75	0.393	
**Elasticity (logarithm)**				
Intervention group	1.42 ± 0.17	1.65 ± 0.23	<0.001	0.035
Control group	1.38 ± 0.14	1.44 ± 0.25	0.325	
**Splenius capitis**				
**Tone (Hz)**				
Intervention group	21.01 ± 1.65	19.07 ± 2.69	0.006	0.094
Control group	21.18 ± 2.47	20.88 ± 3.5	0.655	
**Stiffness (N/m)**				
Intervention group	388.56 ± 48.13	332.22 ± 52.88	<0.001	0.012
Control group	395.24 ± 51.39	385.59 ± 59.32	0.449	
**Elasticity (logarithm)**				
Intervention group	1.49 ± 0.15	1.64 ± 0.23	0.001	0.101
Control group	1.48 ± 0.15	1.53 ± 0.24	0.296	

Data are expressed as mean ± standard deviation. P, *p*-value; T * G, time and group interaction.

**Table 4 ijerph-17-05572-t004:** The changes in cervical and shoulder alignment.

Outcome/Group	Baseline	Two Weeks Post-Treatment	P (Pairwise comparison)	P (T * G)
**Cervical angle (degree)**				
Intervention group	48.06 ± 6.31	50.26 ± 6.22	<0.001	0.016
Control group	50.12 ± 4.62	50.59 ± 5.43	0.352	
**Shoulder angle (degree)**				
Intervention group	60.49 ± 4.57	65.86 ± 4.64	<0.001	<0.001
Control group	62.18 ± 5.49	62.97 ± 6.30	0.293	

Data are expressed as mean ± standard deviation. P, *p*-value; T * G, time and group interaction.

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
