# Peer review of "Thermotherapy Plus Neck Stabilization Exercise for Chronic Nonspecific Neck Pain in Elderly: A Single-Blinded Randomized Controlled Trial"

_ijerph, 2020, doi:10.3390/ijerph17155572_

Round 1

Reviewer 1 Report

The manuscript is well written and the experiment is well conducted. Some comments regarding the method section and a few minor points are indicated. The introduction section addresses the main findings in the literature and the references are adequate. However, the particularity of salt packs and its application and advantages did not be emphasized in the introduction, the novelty and significance of this study cannot be evaluated. 

The purpose of this single-blinded RCT study is to compare the effects of thermotherapy plus neck stabilization exercise versus neck stabilization exercise alone on pain, 15 neck disability, muscle properties, and alignment of the neck and shoulder in elderly with chronic 16 nonspecific neck pain. Pain, PPT for upper-trapezius, levator scapula, and splenius-capitis, NDI, muscle stiffness, and alignment of the treatment group were significantly improved compared to that of the control group. 

The manuscript is well written and the experiment is well conducted. Some comments regarding method section and a few minor points are indicated below. The introduction section addresses the main findings in the literature and the references are adequate. However, the particularity of salt packs and its application and advantages did not be emphasized in the introduction, the novelty and significance of this study cannot be evaluated. 

Introduction

Line 46 Exercise is not necessary a complementary therapy for pain relief. It could be the main and most effective intervention for pain. The article cited does not really support it. Please revise this sentence.

Line 53 It is unclear that how the thermotherapy prevents tissue damage. Please clarify it.

Line 56 Revise as for “relieving nonspecific neck pain.”

Line 57 Wat is “salt pack”? How a salt pack can produce heat? Is it the same as traditional hot pack used in physical therapy? This kind of thermotherapy is not common to most readers. Thus please briefly introduce the salt packs, its history, how it can be applied and its advantages. Authors really need to emphasize this point, otherwise the salt packs is just a heat that is not different from the hot pack.

Methods

Line 72 Revise as”the elders”

Line 86 Stratified according to age or gender ….any specific factors?

Line 88 Whether the post-intervention assessment was performed on the same day immediately after the last intervention session? Please specifically describe this point. Since a salt pack therapy could relieve pain relief and decrease stiffness immediately after application, but its effects may not last long. The significant group differences found in this study could simply the results from immediate effects from salt packs 

Line 96 Please provide an average temperature or range regarding the tolerable temperature

Line 103 VAS during movement is defined as pain incurred by activities of daily living. Are there specific activities assessed? Did authors record which activities can provoke neck pain?

Line 111 How many trials were assessed for PPT?

Line 115 Total score of NDI should be 50, and can be expressed as 100%. Please revise it and make sure your data is correct

Line 122 If the muscle stiffness is measured as the same location as PPT. Did authors randomize the testing sequence to prevent fatigue or over-induce the pain response? Please provide the validity and reliability of the handheld myotonometer for measuring muscle tone, stiffness, and elasticity. The size of probe should also be provided. Also please define muscle tone, stiffness, and elasticity

Line125 How to get the cervical and shoulder alignments? Please provide the information such as using 2D photo or 3D motion analysis or any other method

Line 134 Did authors perform post hoc analysis?

Results

Line 142 Provide the reason for drop out

Line 161 Table2 provide unit for each outcome

Discussion

No discussion on neck and shoulder alignment. How are the associations between neck pain and posture or alignment?

Author Response

Dear Reviewers,

I would like to thank you for providing the opportunity to revise and resubmit the attached manuscript entitled “Thermotherapy Plus Neck Stabilization Exercise for Chronic Nonspecific Neck Pain in Elderly: A Single-Blinded Randomized Controlled Trial” for publication in International Journal of Environmental Research and Public Health.

We deeply appreciate the editorial comments and reviewers’ helpful comments on our manuscript which we ignored. We agreed with the points addressed by the Reviewers. We provide our responses to the Reviewers’ comments. Please review the attached files.

We hope that the revisions we have made are satisfactory. Please inform us if there is anything else that can be done to improve the manuscript.

Thank you for your consideration.

Sincerely,

Hwi-young Cho, PT, PhD,

Department of Physical Therapy, College of Health Science, Gachon University, 191 Hambangmoe-ro, Yeonsu-gu, Incheon 21936, Republic of Korea

Tel: +82-32-820-4560; Fax: +82-32-820-4420; E-mail: hwiyoung@gachon.ac.kr

Suk-Chan Hahm, PT, PhD,

Graduate School of Integrative Medicine, CHA University, CHA Bio Complex, 355, Pangyo-ro, Bundang-gu, Seongnam-si, Kyonggi-do, 13488, Republic of Korea

Tel: +82-31-881-7101; Fax: +82-31-881-7069; E-mail: schahm@cha.ac.kr

Reviewer 2 Report

vv

The authors address an interesting theme, and a well conducted investigation that is based on the study of the same authors - Hahm et al, 2020.

However there are some issues that should be improved:

  • It would be useful to the reader to understand the thermotherapy physiologic tissue effects since it´s the more relevant independent variable.
  • As well more information about the neck stabilization exercises is needed.
  • There is not enough information about the population and studied sample. More information needs to be provided. We only know that the sample has sixty-year-old with chronic neck pain subjects. We don’t understand the context of this sample, the gender characteristics, the job context.
  • The procedures also lack information. What was the experimental set up?
  • When was the thermotherapy done? Before or after the exercises? When and how were the exercises performed? Where did the subjects performed the interventions? At home? Did they received instructions?
  • What were the exercises selected. Maybe some short information would be useful.
  • What are the references for the methodology of the exercises?
  • What are the references and rational for the selected protocol? The references provided don´t present the same methology.
  • What are the limitations in terms of effectivity of this methodology? This should be discussed because the protocol made twice a day more than an hour is a significant time-consuming protocol. Could this intensity impact the results? What about adherence and performance issues?

Author Response

(The authors gave the same response as above.)
